# Immunologic monitoring of cytomegalovirus (CMV) enzyme-linked immune absorbent spot (ELISPOT) for controlling clinically significant CMV infection in pediatric allogeneic hematopoietic stem cell transplant recipients

**Euri Seo**[1,2], **Eun Seok Choi**[1], **Jung Hwa Kim**[1], **Hyery Kim**[1], **Kyung-Nam Koh**[1], **Ho Joon Im**[1], **Jina Lee**[1] *

**1** Department of Pediatrics, Asan Medical Center Children's hospital, University of Ulsan College of Medicine, Seoul, Republic of Korea, **2** Department of Pediatrics, Dongguk University Ilsan Hospital, Dongguk University College of Medicine, Goyang, Republic of Korea

* entier@amc.seoul.kr

## Abstract

The dynamics of recovery of cytomegalovirus (CMV)-specific cell-mediated immunity (CMI) and its impact on controlling clinically significant CMV infections following hematopoietic stem cell transplant (HSCT) are rarely reported in pediatric HSCT recipients. In this study, dynamics of recovery of CMV-specific CMI and its clinical significance in controlling CMV viremia and clinically significant CMV infections were assessed in pediatric allogeneic HSCT recipients. All subjects underwent CMV pp65- and IE1-specific enzyme-linked immune absorbent spot (ELISPOT) assays just before transplantation and then monthly until the detection of CMV-specific CMI with $\geq$ 5 spot-forming cells (SFC) / $2.0 \times 10^5$ cells. Clinically significant CMV infections were defined as CMV diseases, prolonged CMV infections, recurrent CMV infections or late onset CMV infections. Among 52 recipients, 88.5% of recipients recovered CMV-specific CMI with $\geq$ 5 SFC/ $2.0 \times 10^5$ cells at a median of 34 days (interquartile range [IQR]: 29–95 days) following HSCT, 55.8% at 30 days following HSCT, and 73.1% at 90 days following HSCT. The presence of CMV-specific CMI before HSCT was the significant factors for the reconstitution of CMV specific CMI after HSCT (adjusted odds ratio [aOR] = 13.33; 95% confidence interval [CI] = 1.21–142.86). After HSCT, 30 recipients experienced CMV viremia, of which 20 were clinically significant CMV infections. The full recovery of CMV-specific CMI with $\geq$ 50 SFC / $2.0 \times 10^5$ cells after HSCT was the protective factor for the development of clinically significant CMV infections (aOR = 0.13; 95% CI = 0.22–0.71). In the haploidentical HSCT recipients, 82.1% recovered CMV-specific CMI at a median of 65 days after HSCT (IQR: 34–118 days) with a tendency to recover their CMV-specific CMI later than did those from non-haploidentical donors (65 days vs. 30 days; $P = 0.001$). Clinically significant CMV infections tended to occur more frequently in the haploidentical HSCT recipients compared to those with matched donor HSCT (46.4% vs. 29.2%; $P = 0.205$). The full recovery of CMV-specific CMI with $\geq$ 50 SFC/$2.0 \times 10^5$ cells after HSCT also lowered the risk of development of clinically significant CMV infections

**Data Availability Statement:** All relevant data are within the manuscript and its Supporting Information files.

**Funding:** The author(s) received no specific funding for this work.

**Competing interests:** The authors have declared that no competing interests exist.

(aOR = 0.08; 95% CI = 0.01–0.90). However, transplantation from haploidentical donors was a significant risk factor hampering recovery of CMV-specific CMI (aOR = 0.08; 95% CI = 0.01–0.86) and full recovery of CMV-specific CMI (aOR = 0.05; 95% CI = 0.01–0.50). Pre-transplant CMV-specific CMI influenced the recovery of CMV-specific CMI, and the full recovery of CMV-specific CMI could be a surrogate marker for preventing clinically significant CMV infections in pediatric HSCT recipients. Immunologic monitoring using ELISPOT assay before and after HSCT helps in identifying patients with a high risk of CMV infection and in controlling CMV infection.

## Introduction

Cytomegalovirus (CMV) infection and CMV disease are associated with high morbidity and mortality in patients undergoing hematopoietic stem cell transplant (HSCT) despite the use of antiviral prophylaxis or preemptive therapy [1–3]. The recovery of CMV-specific cell-mediated immunity (CMI) following HSCT plays an important role in treating CMV infection and preventing CMV disease both in adults and children [4, 5], and the high CMV T-cell response after HSCT is a protective factor against CMV reactivation [6]. In patients with CMV-specific CMI, the early discontinuation of antiviral use for treating CMV infection and spontaneous clearance of low-level viremia without antiviral treatment are possible [5, 7]. In studies of pediatric HSCT recipients, CMV-specific CMI reconstitution lowered the risk of CMV infection [8, 9].

Several studies have been performed to measure the recovery of CMV-specific CMI after HSCT, and they have reported that enzyme-linked immune absorbent spot (ELISPOT) assay is useful for measuring CMV-specific CMI [6, 8, 10]. Both CD4+ and CD8+ T cells are necessary for protecting against CMV infection. The CMV ELISPOT assay allows the evaluation of CD4+ and CD8+ T-cell immunity by an *ex vivo* production and release of interferon gamma (IFN- γ) via the reaction of both CD4+ and CD8+ T cells with CMV antigens. It has become commercially available in the United States, Europe, and Korea.

Recently, as it has become difficult to find suitable human leukocyte antigen (HLA)-matched donors, HSCT from HLA-haploidentical donors has been practiced. In HSCT from HLA-haploidentical donors, a combination of the infusion of high doses of hematopoietic stem cells and profound T-cell depletion are applied to prevent graft rejection associated with the high degree of HLA disparity between the donor and recipient and to prevent the development of life-threatening graft-versus-host disease (GvHD) [11]. For these reasons, fatal infections including serious CMV infection can occur with increased frequency in recipients given a T-cell–depleted transplant from an HLA disparate relative. A faster recovery of CMV-specific T-cell responses is associated with a reduced CMV reactivation incidence in HSCT from haploidentical donors [12].

The purpose of this study was to evaluate the dynamics of recovery of CMV-specific CMI and to determine the clinical significance of CMV-specific CMI in controlling CMV viremia and clinically significant CMV infections in pediatric HSCT recipients, especially focused on HSCT recipients from haploidentical donors.

## Materials and methods

The study was approved by an institutional review board of Asan medical center (IRB No. 2016–0220). We obtained written consent.

## Patients

We prospectively enrolled pediatric patients $\leq$ 18 years old who underwent allogeneic HSCT with at least 12 months of follow up at Asan Medical Center Children's Hospital between March 2016 and June 2017. The following cases were excluded from this study: (1) a refusal of written informed consent, 2) pre-HSCT CMV serostatus of a negative donor (D-) and negative recipient (R-), 3) death within one month after HSCT, 4) engraftment failure within one month after transplantation, or 5) relapse of underlying disease during follow up. The following case information was abstracted from electronic medical records: age at time of transplantation; gender; underlying disease leading to HSCT; types of donor (i.e., matched sibling, matched unrelated, or haploidentical); conditioning regimen (i.e., myeloablation vs. non-myeloablation); CMV serostatus before transplantation; previous history of CMV viremia before transplantation; acute and chronic GvHD following HSCT; recurrence of the underlying disease; and death. The CMV serostatus at the time of HSCT was determined by the presence of CMV immunoglobulin G (IgG). Recipients < 1 year old were considered as seronegative even if they had CMV IgG, unless CMV viremia was detected; this is because maternal CMV IgG can cross the placenta and be detected until about 12 months of age.

## Evaluation of CMV-specific CMI

A peripheral venous blood sample up to 5 mL was collected from each patient for detecting CMV specific T cells producing IFN-$\gamma$ by ELISPOT assay (T-track CMV, Regensburg, Germany). Peripheral blood mononuclear cells (PBMCs) were separated and collected. The collected cells were resuspended at $2.0 \times 10^6$ cells/mL and placed in wells that were pre-coated with anti-human IFN-$\gamma$ antibody ($2.0 \times 10^5$ cells/well). The PBMCs were stimulated with phytohemagglutinin (positive control), pp65, IE-1, and medium only (negative control), and were then incubated for 12 hours. The resulting spots were counted by a skilled researcher with an automated microscope. Background counts, obtained in the negative control wells, were subtracted. The results were expressed as spot-forming cells (SFC) / $2.0 \times 10^5$ cells. The recovery of CMV-specific CMI was defined as the detection of $\geq$ 5 SFC/$2.0 \times 10^5$ cells for either pp65 or IE-1 at least 2 consecutive times, and full recovery of CMV-specific CMI was defined as a cut off value of $\geq$ 50 SFC / $2.0 \times 10^5$ cells.

CMV-specific CMI monitoring was performed according to the following protocol: once before the start of conditioning regimens, monthly for the first six months following HSCT, and then every three months until CMV-specific CMI was recovered. If CMV-specific CMI was not recovered after 1 year following HSCT, it was monitored up to 18 months with an interval of 6 months. If CMV-specific CMI was shown to be recovered at two consecutive evaluations, the monitoring was terminated.

## CMV viral load monitoring and clinically significant CMV infection

For the quantitative analysis of CMV viremia, the patients' whole blood was collected and analyzed using quantitative CMV polymerase chain reaction (PCR) with CMV antigenemia. The CMV pp65 antigenemia or PCR were monitored weekly from infusion to day 100 after HSCT and then monthly between 100 days and 1 year after HSCT. The CMV antigenemia assay was performed as previously described, with counts expressed as positive cells per 200,000 leukocytes [13]. CMV DNA quantitation was performed with a Qiagen Artus CMV Rotor-Gene Q (RGQ) MDx kit (Qiagen, Doncaster, Australia) on an RGQ platform (Qiagen, Doncaster, Australia) following DNA extraction with a NucliSens easyMAG nucleic acid extraction system (bioMerieux, Lyon, France). The linear range of the CMV PCR used for the quantification analysis in this study was 2.69–7.00 log copies/mL (489–50,000,000 copies/mL), and the

 

detection limit was 2.69 log copies/mL (489 copies/mL). CMV viremia was defined as the CMV PCR titer being above 2.69 log copies per 1 mL of whole blood (489 copies/mL) even once, or the CMV antigenemia being above 1 cell/200,000 cell in twice consecutively.

CMV disease was defined according to the definition suggested by Ljungman et al. [14] Prolonged CMV viremia was defined as CMV viremia lasting for ≥ 2 weeks, and recurrent CMV viremia was defined as new CMV viremia in a patient with a previous history of CMV viremia in whom the virus had not been detected for an interval of at least four weeks. Late-onset CMV viremia was defined as CMV viremia that occurred after 100 days of HSCT. In this study, clinically significant CMV infection was defined as the presence of at least one of the following conditions: prolonged, recurrent, or late-onset CMV viremia, or CMV disease.

## CMV prophylaxis and preemptive CMV therapy

All recipients who received a HSCT from a matched sibling or unrelated donor did not receive CMV prophylaxis irrespective of CMV serostatus. Instead, as a preemptive treatment, blood CMV viral loads were monitored regularly, and antiviral treatment with either ganciclovir (5 mg/kg/dose every 12 hours) or foscarnet (60 mg/kg/dose every 12 hours) was performed when CMV viremia was detected. This preemptive treatment was discontinued following 2 consecutive negative PCR results.

In recipients with haploidentical HSCT, ganciclovir (5 mg/kg/dose every 24 hours) or foscarnet (60 mg/kg/dose every 24 hours) was used for CMV prophylaxis up to 100 days after engraftment. After that, oral acyclovir was used prophylactically until the first year of transplantation. For CMV viremia, anti-CMV agents were administered for therapeutic purposes.

## Ethical considerations

This study protocol was approved by an institutional review board of our hospital (IRB No. 2016–0220). Informed written consent for participation in the study was obtained from recipients' parents when deciding the transplantation date. They were told about the benefits and risks of participating in the study and the protection of the participants' privacy.

Search and selection of each donor for HSCT was determined in the order of matched sibling donor, matched unrelated donor, and haploidentical donor. When it was confirmed that the donor's hematopoietic stem cells were suitable for the recipient, consent for donation was then obtained from the donor. In the case of HSCT from matched sibling donor and haploidentical donor, the donor agreed to the donation after understanding how to collect the stem cells and the potential risk of collecting the stem cells. However, when the donor was a child, the consent was received from their parents. Matched unrelated donor in this study was selected among people enrolled in Korea Marrow Donor Program who were willing to donate their stem cells voluntarily. Every donor provided their stem cells without any monetary compensation. Collection of hematopoietic stem cells from donors was also performed from March 2016 to June 2017.

## Statistical analysis

Demographic and clinical characteristics were compared using the Mann-Whitney U test for continuous variables and the χ2 test or Fisher's exact test for categorical variables. Univariate or multivariate regression analyses were used to elucidate the factors for related to the reconstitution of CMV-specific CMI or the development of CMV viremia and clinically significant CMV infection. *P* values < 0.05 were considered statistically significant. Statistical analyses were performed using SPSS version 20.0 (IBM Software, Chicago, IL, USA).

# Results

## Study population

Between March 2016 and June 2017, a total of 52 allogeneic HSCT recipients with a median age of 10.2 years (inter-quartile range [IQR], 3.2–14.8 years) were enrolled (Table 1). Of the total study population, 57.7% (30/52) of the patients underwent HSCT because of underlying malignancies and 71.2% (37/52) received non-myeloablative conditioning. All patients received peripheral blood stem cells (PBSCs) as an allogeneic graft, either from an HLA-matched sibling (n = 12; 23.1%), an HLA-matched or one mismatched unrelated donor (n = 12; 23.1%), or a

**Table 1. Demographic and clinical characteristics of total 52 allogeneic HSCT recipients.**

| Characteristics | Number of cases (%) |
|---|---|
| **Male: Female** | 34:18 |
| **Median age at HSCT, years (IQR)** | 10.2 (3.2–14.8) |
| **Underlying disease** | |
| **Malignancy** | 30 (57.7) |
| **Acute myeloid leukemia** | 16 (30.8) |
| **Acute lymphoblastic leukemia** | 7 (13.5) |
| **Solid tumor** | 3 (5.8) |
| **Hemophagocytic lymphohistiocytosis** | 4 (7.7) |
| **Non-malignancy** | 22 (42.3) |
| **Aplastic anemia** | 15 (28.8) |
| **Refractory cytopenia of childhood** | 2 (3.9) |
| **Others [a]** | 5 (9.6) |
| **Donor type** | |
| **Sibling** | 12 (23.1) |
| **Unrelated [b]** | 12 (23.1) |
| **Haploidentical** | 28 (53.8) |
| **Use of GvHD prophylaxis [c]** | 24 (46.2) |
| **Conditioning regimen** | |
| **Myeloablation** | 15 (29.8) |
| **Non-myeloablation** | 37 (71.2) |
| **CMV serostatus (IgG)** | |
| **Donor +/Recipient +** | 45 (86.5) |
| **Donor −/Recipient +** | 4 (7.7) |
| **Donor +/Recipient −** | 3 (5.8) |
| **CMV CMI, pre-HSCT [d]** | |
| **Positive** | 22 (42.3) |
| **Negative** | 20 (38.5) |
| **Indeterminate** | 9 (17.3) |
| **History of CMV viremia before HSCT** | 2 (3.8) |

Abbreviations: CMV, cytomegalovirus; CMI, cell mediated immunity; HSCT, hematopoietic stem cell transplantation; GvHD, graft-versus-host disease; IQR, inter-quartile range.

[a] Others include Wiskott-Aldrich syndrome (n = 2), adrenoleukodystrophy (n = 1), chronic granulomatous disease (n = 1) and interleukin-10 deficiency (n = 1).

[b] Four of 12 patients received graft from human leukocyte antigen (HLA) DR 1 allele mismatched donor.

[c] GvHD prophylaxis regimen: Cisplatin combined with methotrexate or mycophenolate mofetil.

[d] Presence of pre-HSCT CMV specific CMI was defined as detection of $\geq$ 5 SFC/2.0 x $10^5$ cells for either pp65 or IE-1 before start of conditioning regimens.

haploidentical donor (n = 28; 53.8%). None of the haploidentical HSCT recipients received GvHD prophylaxis after HSCT. The CMV D+/R+ group was the most prevalent at 86.5%, followed by D−/R+ at 7.7% and D+/R− at 5.8%. Pre-HSCT CMV-specific CMI was present in 22 recipients (42.3%); however, it was indeterminate in 9 recipients and absent in 21 recipients.

## Clinical outcomes including CMV infections following HSCT

Overall, 30 recipients (57.7%) experienced CMV viremia at a median of 32 days (IQR: 20–47 days) following HSCT. Clinically significant CMV infection occurred in 20 recipients including CMV diseases (n = 5, all of which were retinitis), prolonged CMV viremia (n = 18), and recurrent CMV viremia (n = 7). Late-onset CMV viremia occurred in 6 recipients, all of whom received haploidentical HSCT.

Acute and chronic GvHD occurred in 36.5% (19/52) and 17.3% (9/52), respectively. The fatality rate among all HSCT recipients was 7.7% (4/52). Among three fatal cases who had CMV viremia at the time of death, the causes of death were as follows: *S. aureus* bacteremia at 4 months following HSCT, and transplantation-associated thrombotic microangiopathy at 1 month and at 7 months after HSCT, respectively. The one remaining fatal case was associated with uncontrolled gastrointestinal bleeding at 7 months without CMV viremia after HSCT.

There was no significant difference in the incidence of CMV viremia among haploidentical HSCT recipients and non-haploidentical HSCT (57.1% and 58.3%, respectively; *P* = 0.931) (Table 2). Among the total 28 haploidentical HSCT recipients whose median age was 8.4 years (IQR: 3.0–13.0 years), 16 recipients developed CMV viremia at a median of 40 days post HSCT (IQR: 28–75.5 days). Overall, 81.3% (13/16) of haploidentical HSCT recipients who experienced CMV viremia after HSCT were diagnosed with clinically significant CMV infection, 4 recipients with CMV retinitis, 11 with prolonged CMV viremia, 6 with recurrent CMV viremia, and 6 with late-onset CMV viremia. Clinically significant CMV infection tended to occur more frequently in recipients with haploidentical HSCT compared to those with matched donor HSCT (*P* = 0.205).

**Table 2. Clinical outcomes including CMV infections according to the type of HSCT.**

| Variables | Haploidentical HSCT (n = 28) | Full matched or one mismatched HSCT (n = 24) | *P*-value |
|---|---|---|---|
| **CMV viremia (n = 30)** | 16 (57.1) [a] | 14 (58.3) | 0.931 |
| **Clinically significant CMV infection (n = 20)** | 13 (46.4) | 7 (29.2) | 0.205 |
| CMV disease (n = 5) | 4 (14.3) | 1 (4.2) | 0.245 |
| Prolonged CMV viremia (n = 18) | 11 (39.3) | 7 (29.2) | 0.446 |
| Recurrent CMV viremia (n = 7) | 6 (21.4) | 1 (4.2) | 0.101 |
| Late onset CMV viremia (n = 6) | 6 (21.4) | 0 (0) | 0.998 |
| **Death (n = 4)** | 3 (10.7) | 1 (4.2) [b] | 0.394 |
| CMV viremia associated death [c] | 3 (0.0) [c] | 0 (0) | |
| **Acute GvHD [d] (n = 19)** | 11 (39.3) | 8 (33.3) | 0.657 |
| **Chronic GvHD (n = 9)** | 3 (10.7) | 6 (25.0) | 0.186 |

Abbreviations: CMV, cytomegalovirus; GvHD, graft-versus-host disease; HSCT, hematopoietic stem cell transplantation.

[a] Values are number (%) unless otherwise indicated.

[b] One fatal case was associated with gastrointestinal bleeding at 7 months after HSCT.

[c] Along with CMV viremia, one fatal case was associated with *S. aureus* bacteremia at 4 months after HSCT, 2 from transplantation associated thrombotic microangiopathy at 1 month and 7 months after HSCT, respectively.

[d] The grade of acute GVHD is as follows; grade 1 (n = 2), grade 2 (n = 12), grade 3 (n = 4), and grade 4 (n = 1).

## Factors associated with clinically significant CMV infections following HSCT

In univariate and multivariate analyses conducted to elucidate the factors associated with clinically significant CMV infections after pediatric allogeneic HSCT, the full recovery of CMV-specific CMI with $\geq 50$ SFC/$2.0 \times 10^5$ cells was associated with a reduced risk of the development of clinically significant CMV infections (adjusted odd ratio [aOR] = 0.13; 95% confidence interval [CI], 0.02–0.71) (Table 3). When considering only haploidentical HSCT recipients, the full recovery of CMV-specific CMI with $\geq 50$ SFC/$2.0 \times 10^5$ cells following HSCT also lowered the risk of development of clinically significant CMV infections (aOR = 0.08; 95% CI = 0.01–0.90).

However, neither the preexistence nor recovery of CMV-specific CMI following HSCT by $\geq 5$ SFC/$2.0 \times 10^5$ cells had a direct effect on the development of clinically significant CMV infections following HSCT (aOR = 0.84 and 95% CI = 0.15–4.61; aOR = 8.75 and 95% CI = 0.90–84.80, respectively). In addition, haploidentical HSCT itself (aOR = 1.58; 95% CI = 0.26–9.76) and CMV serostatus were not a statistically significant risk factor for clinically significant CMV infections following HSCT.

## Reconstitution of CMV-specific CMI

Overall, 46 recipients (88.5%) recovered CMV-specific CMI during the follow-up period $\geq 1$ year after HSCT. The recovery and full recovery of CMV-specific CMI following HSCT occurred at a median of 34 days (IQR: 29–95 days) and 48 days (IQR: 29–90 days), respectively (Fig 1). CMV-specific immunity determined by pp65 antigen (at a median of 34 days following HSCT; IQR: 29–96 days) was restored earlier than that determined by IE-1 (at a median of 60.5 days; IQR: 30–108 days). The cumulative recovery of CMV-specific CMI according to time period was as follows: 55.8% (29/52) at 30 days and 73.1% (38/52) at 90 days following HSCT. After 3 months following HSCT, CMV-specific CMI was recovered at 4 months (n = 5), 5 months (n = 2) and 6 months (n = 1).

**Table 3. Factors associated with clinically significant CMV infections following HSCT.**

| Variables | OR (95% CI) | Adjusted OR (95% CI) |
|---|---|---|
| **Age at HSCT** | 1.00 (0.99–1.01) | 1.00 (0.99–1.01) |
| **Underlying disease of malignancy** | 0.43 (0.14–1.35) | 0.61 (0.15–2.51) |
| **Haploidentical HSCT** | 2.1 (0.67–6.67) | 1.58 (0.26–9.76) |
| **Myeloablative conditioning** | 0.48 (0.13–1.78) | 0.48 (0.12–1.98) |
| **CMV serostatus** | | |
| Donor +/Recipient + | 1.00 | 1.00 |
| Donor +/Recipient − | 3.63 (0.31–43.15) | 7.66 (0.34–173.72) |
| Donor −/Recipient + | 1.81 (0.23–14.12) | 3.37 (0.28–41.10) |
| **Preexistence of CMV specific CMI at HSCT** | 1.39 (0.40–4.80) | 0.84 (0.15–4.61) |
| **CMV viremia before HSCT** | 1.63 (0.10–27.65) | 2.98 (0.14–62.73) |
| **CMV specific CMI after HSCT** | | |
| $\geq 50$ SFC/$2.0 \times 10^5$ cells | 0.34 (0.10–1.16) | 0.13 (0.02–0.71) |
| $\geq 20$ SFC/$2.0 \times 10^5$ cells | 0.54 (0.15–1.98) | 0.14 (0.01–1.32) |
| $\geq 5$ SFC/$2.0 \times 10^5$ cells | 1.67 (0.29–9.54) | 8.75 (0.90–84.80) |

Abbreviations CMV, cytomegalovirus; CI, confidence interval; CMI, cell mediated immunity; ELISPOT, enzyme-linked immunospot; HSCT, hematopoietic stem cell transplantation; GvHD, graft-versus-host disease; OR, odds ratio.

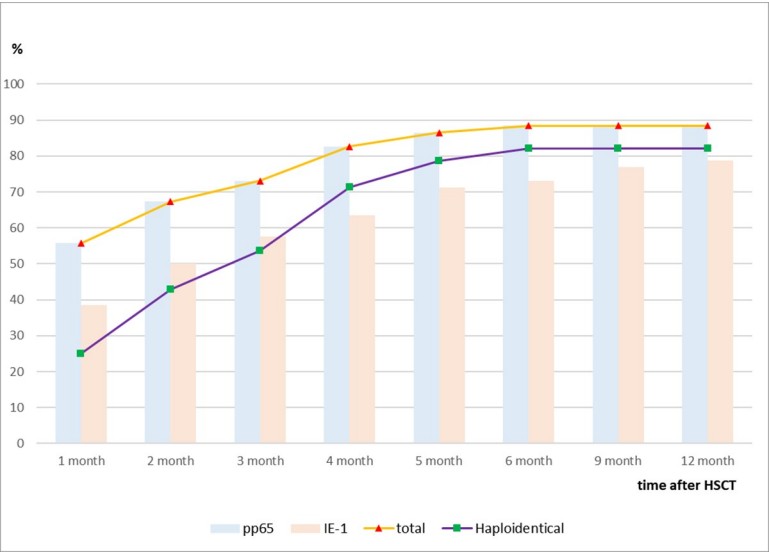

**Fig 1. Cumulative rate of recovery of CMV-specific CMI over time after HSCT.** CMV-specific immunity determined by pp65 antigen (blue bar) was restored earlier than that determined by IE-1 (pink bar). Compared to overall HSCT recipients (red triangle), recipients from haploidentical donors (green square) tended to recover their CMV-specific CMI later.

A total of 6 recipients did not recover CMV-specific CMI within one year following HSCT: 1 recipient died of *S. aureus* bacteremia at 4 months following HSCT; 1 patient (who underwent lung transplantation at 10 months after HSCT) recovered CMV-specific CMI at 18 months following HSCT; and the remaining 4 did not recover the CMV-specific CMI even after 18 months of HSCT.

In the subgroup analysis of haploidentical HSCT recipients, 82.1% (23/28) recovered CMV-specific CMI at a median of 65 days after HSCT (IQR: 34–118 days). Recipients from haploidentical donors tended to recover their CMV-specific CMI later than did those from non-haploidentical donors (65 days vs. 30 days; $P = 0.001$).

### Factors associated with the recovery of CMV-specific CMI following HSCT

The recovery of CMV-specific CMI by $\geq 5$ SFC/$2.0 \times 10^5$ cells following HSCT was significantly affected by the preexistence of CMV-specific CMI at the time of HSCT (aOR = 13.33; 95% CI = 1.21–142.86) (Table 4). Transplantation from a haploidentical donor was a significant risk factor for hampered the recovery of CMV-specific CMI (aOR = 0.08; 95% CI = 0.01–0.86).

Considering the full recovery of CMV-specific CMI with $\geq 50$ SFC/$2.0 \times 10^5$ cells, haploidentical HSCT also had a negative effect on the recovery of CMV-specific CMI (aOR = 0.05; 95% CI = 0.01–0.50). In addition, CMV serostatus in D−/R+ cases exerted a negative influence on the full recovery of CMV-specific CMI (aOR = 0.04; 95% CI = 0.00–0.80). However, the preexistence of CMV-specific CMI at the time of HSCT did not have a statistically significant effect on the full recovery of CMV-specific CMI following HSCT (aOR = 3.75; 95% CI = 0.64–22.22).

## Discussion

Our study suggests that the presence of pre-HSCT CMV-specific CMI $\geq 5$ SFC/$2.0 \times 10^5$ cells is a significant predictive factor of the recovery of CMV-specific CMI following HSCT. In

**Table 4. Factors associated with the recovery of CMV specific CMI following HSCT.**

| Variables | Cut-off value of CMV specific CMI by $\geq 5$ SFC/$2.0 \times 10^5$ cells | | Cut-off value of CMV specific CMI by $\geq 50$ SFC/$2.0 \times 10^5$ cells | |
|---|---|---|---|---|
| | OR (95% CI) | aOR (95% CI) | OR (95% CI) | aOR (95% CI) |
| **Age at HSCT** | 1.01 (1.00–1.02) | 1.01 (0.99–1.03) | 1.00 (0.99–1.01) | 1.00 (0.99–1.01). |
| **Underlying disease of malignancy** | 1.03 (0.21–5.13) | 0.58 (0.06–5.78). | 2.28 (0.69–7.55) | 1.79 (0.40–7.97) |
| **HSCT from haploidentical donor** | 0.16 (0.02–1.43) | 0.08 (0.01–0.86) | 0.17 (0.04–0.68) | 0.05 (0.01–0.50) |
| **CMV serostatus** | | | | |
| Donor +/Recipient + | Reference | | Reference | |
| Donor +/Recipient − | $2.49 \times 10^8$ (0.00-∞) | $8.05 \times 10^8$ (0.00-∞) | 0.81 (0.07–9.76) | 0.29 (0.01–7.69) |
| Donor −/Recipient + | 0.46 (0.04–5.20) | 0.00 (0.00-∞). | 0.41 (0.05–3.20) | 0.04 (0.00–0.80) |
| **Presence of pre-HSCT CMI** | 7.33 (0.78–69.24) | 13.33 (1.21–142.86) | 1.53 (0.41–5.64) | 3.75 (0.64–22.22) |

Abbreviations: CMV, cytomegalovirus; CMI, cell mediated immunity; HSCT, hematopoietic stem cell transplantation; GvHD, graft-versus-host disease; CI, confidence interval; N.A., not applicable; OR, odds ratio; aOR, adjusted odd ratio.

addition, haploidentical donors and CMV serostatus in D−/R+ cases were significant risk factors impeding the full recovery of CMV-specific CMI $\geq 50$ SFC/$2.0 \times 10^5$ cells. Furthermore, the full recovery of CMV-specific CMI could reduce the risk of the development of clinically significant CMV infections in pediatric allogeneic HSCT recipients. In recipients who have undergone haploidentical HSCT, CMV-specific CMI tended to recover late and clinically significant CMV infections tended to occur more frequently.

Although the methods of measuring CMV-specific CMI and the cut-off value of its recovery following HSCT vary from study to study, several studies have been reported including the pediatric populations. A previous study by Merindol et al. showed reconstitution of CMV-specific CMI with $\geq 10$ SFC/$2.0 \times 10^5$ cells assessed by ELISPOT was observed in 3.57% of pediatric HSCT recipient from cord blood stem cell at 2 and 3 months posttransplant and in 16.3% at 12 months, respectively; at 36 months, 33.5% of recipients had developed CMV-specific CMI [9]. Other study by Paouri et al. reported that approximately 40% of pediatric patients receiving HSCT recovered CMV-specific immunity at an average of 82 days after transplantation, and a total of 46% recovered it at 1 year after transplantation, when CMV-specific immunity was monitored using the ELISA based Quantiferon-CMV assay and recovery was defined as when the Quantiferon result was $\geq 0.2$ IU/mL [15]. In our study, about 90% of pediatric HSCT recipients recovered CMV-specific CMI with $\geq 5$ SFC/$2.0 \times 10^5$ cells at 1 year after HSCT, and the median time for recovery was 1 month after HSCT. These findings might be a combined result of differences in the way detecting CMV-specific T cells producing IFN-γ, differences in the CMV seroprevalence, graft source, conditioning regimens, and use of immunosuppressive agents for preventing GvHD between this and other study. In the haploidentical HSCT recipients, 82.1% recovered CMV-specific CMI at 1 year after HSCT at a median of 2 months following HSCT. This result is the same as other study in young patients who received haploidentical HSCT [16]; about 82% of recipients showed the recovery of CMV-specific T-cell immunity at a median of 72 days after transplantation when subjects with a CMV-specific CD4$^+$ or CD8$^+$ T cell count of > 0.4 cells/uL were considered to be immune. Compared to the recipients with matched HSCT, the haploidentical HSCT recipients experienced a significantly delay in CMV-specific T-cell reconstitution. It has been suggested that these differences in immune recovery according to donor types are associated with the number of T cells in the graft and the age of the recipients [16].

Even though the recovery of CMV-specific CMI was slower by one month in haploidentical HSCT recipients compared to full-matched donors, haploidentical HSCT itself was not a

definite risk factor for CMV viremia or clinically significant CMV infections after HSCT. However, among those with CMV viremia, the proportion of recipients who developed clinically serious CMV infections tended to be higher in the haploidentical HSCT group compared to the group that underwent matched HSCT. The full recovery of CMV-specific CMI after HSCT had a protective effect on the development of clinically significant CMV infections. More than 40 SFC/$2.0 \times 10^5$ for IE-1 and more than 80 SFC/$2.0 \times 10^5$ for pp65 was protective against CMV infection and progression to clinically significant CMV infections in adult allogeneic HSCT recipients [6, 17]. In pediatric allogeneic HSCT, more than 80 SFC/$2.0 \times 10^5$ cells can be used as a clinical threshold for showing a protective effect against CMV viremia, whereas less than 20 SFC/$2.0 \times 10^5$ cells as potential risk factor for developing CMV viremia [8]. Furthermore, no CMV-related events have been observed in pediatric recipients who underwent umbilical cord blood transplantation and had restored CMV-specific CMI $\geq$ 30 SFC/$2.0 \times 10^5$ following HSCT, in which study the recovery of CMV-specific CMI was defined as more than 10 SFC/$2.0 \times 10^5$ [9]. These findings suggest that the sufficient recovery of CMV-specific CMI following HSCT is an important factor to minimize morbidity and mortality associated with CMV infections in pediatric HSCT recipients.

Comprehensive studies on the association between pre-HSCT CMV-specific CMI and the recovery of CMV-specific CMI after HSCT are lacking. We found that the preexistence of CMV-specific CMI had a positive effect on the recovery of CMV-specific CMI, even though there was no statistically significant impact on the full recovery of CMV-specific CMI following HSCT in our study. Some previous studies reported the CMV seropositivity of the recipient before HSCT was strongly associated with the CMV-specific immunity after HSCT [18, 19]. Pre-transplantation CMV infection of the recipients expressed as CMV seropositivity may drive CMV-specific CMI after HSCT by acting as a booster for donor-derived, antigen-experienced T cells [20, 21]. Although the CMV seropositivity before HSCT may be false positive due to passive IgG administration with blood products or immunoglobulin infusion, the presence of CMV-specific CMI evaluated by ELISPOT actually reflects that there was a CMV infection before transplantation.

Of the recently published data, a study by Bae et al. reported that a high CMV-specific T-cell response following HSCT was not associated with pre-HSCT CMV-specific T-cell response, which was similar to our results [22]. The presence of pre-HSCT CMV-specific CMI did not have a statistically significant effect on the full recovery of post-HSCT CMV-specific CMI. These results suggest that the presence of pre-HSCT CMV-specific CMI can predict the possibility of recovering CMV-specific CMI following HSCT but cannot be used as a surrogate marker for predicting clinically serious CMV infection in pediatric allogeneic HSCT recipients. Further studies are needed to confirm these findings.

In this study, CMV serostatus in D−/R+ cases had a negative effect on the full recovery of CMV-specific CMI. This result is also in agreement with some previous studies showing better CMV-specific T-cell reconstitution in D+/R+ patients compared with D−/R+ patients. More CMV-specific T-cell responses were detected at day 100 in D+/R+ patients than in D−/R + patients (78% vs 58%) [23]. This was also observed in studies of both T cell depleted HSCT and T-cell replete HSCT [24–26]. Transfer of both naïve and memory CMV-specific T cells present in the graft from CMV seropositive donors may provide CMV-specific cellular immunity to the recipients.

Although it is well known that haploidentical HSCT generally shows profound and prolonged immune incompetence because of aggressive GvHD prophylaxis [27, 28], our center applies the TCRαβ T-cell/CD19+ depletion approach for all haploidentical HSCT, without the use of immunosuppressive agents for GvHD prophylaxis and CMV prophylaxis following HSCT. There were no studies on the recovery of CMV-specific CMI in children who

underwent haploidentical HSCT using our protocol, and in this study, haploidentical HSCT had a negative effect on the recovery of CMV-specific CMI after HSCT. So, regardless of GvHD prophylaxis regimens, pediatric recipients who underwent haploidenitical HSCT should be cautiously monitored in terms of CMV-specific CMI recovery and the emergence of serious CMV infections.

This study has several limitations. First, it is difficult to discriminate the significance of CMV serostatus in D–/R+ and D+/R–cases to the development of serious CMV infections or the recovery of CMV-specific CMI because of the small sample size and the fact that most of the cases of CMV serostatus were D+/R+. However, this study could suggest that CMV D–/R + cases could be a surrogate marker for the interruption of the full recovery of CMV-specific CMI without having a direct effect on clinically significant CMV infections, and suggesting that clinicians should pay more attention to the CMV D–/R+ group. Second, ELISPOT assay results might differ in terms of inter-examiner or inter-lab calculations of the number of spot counts [29, 30]. In addition, it is difficult to present an appropriate universal cut off value for preventing clinically significant CMV infections. In this study, the values of $5 \text{ SFC}/2.0 \times 10^5$ and $50 \text{ SFC}/2.0 \times 10^5$ were used as our own criteria for general and complete CMV-specific CMI recovery, respectively. Third, in this study, heterogenous patient groups were included in terms of underlying disease leading to HSCT, types of donor, conditioning regimen, CMV serostatus before transplantation, and pre-HSCT CMV-specific CMI, etc. The lack of number of subjects included in this study made further analysis of CMV-specific CMV recovery or CMV infection by each subgroup difficult. However, the reconstitution of T-cell immune response —including CMV specific T-cell immunity—is affected by donor source, degree of match and conditioning regimen [31]. So, an analysis that considers the interaction between these various factors together may be used more meaningfully in the actual clinical field.

In conclusion, the pre-existence of CMV-specific CMI, donor type, and CMV serostatus before HSCT were the major determinants of general or complete CMV-specific CMI recovery after HSCT, and complete recovery of CMV-specific CMI was important for the prevention of clinically significant CMV infection. Immunologic monitoring using ELISPOT assay before and after HSCT helps in identifying patients with a high risk for CMV infection and in minimizing CMV-related morbidity and mortality. Application of this approach could not only reduce the duration of CMV viral load monitoring but also reduce the use of antiviral agents used to control CMV viremia and clinically significant CMV infection and the associated side effects of use of antiviral agents. Further studies among pediatric patients with various serologies and donor types are needed.

## Supporting information

**S1 Data.**
(ZIP)

**S2 Data.**
(ZIP)

## Author Contributions

**Conceptualization:** Jina Lee.

**Data curation:** Euri Seo.

**Formal analysis:** Euri Seo.

**Investigation:** Eun Seok Choi, Jung Hwa Kim, Hyery Kim, Kyung-Nam Koh, Ho Joon Im.

**Methodology:** Jina Lee.

**Supervision:** Jina Lee.

**Writing – original draft:** Euri Seo.

**Writing – review & editing:** Jina Lee.

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
