## [Decision Letter · Decision Letter 0]

21 Oct 2020

PONE-D-20-22050

Immunologic monitoring of cytomegalovirus (CMV) enzyme-linked immune absorbent spot (ELISPOT) for controlling clinically significant CMV infection in pediatric allogeneic hematopoietic stem cell transplant recipients

PLOS ONE

Dear Dr. Lee,

Thank you for submitting your manuscript to PLOS ONE. After careful consideration, we feel that it has merit but does not fully meet PLOS ONE’s publication criteria as it currently stands. Therefore, we invite you to submit a revised version of the manuscript that addresses the points raised during the review process.

We have now received reports from two referees of your manuscript, as agree with reviewers comments raised a few concerns about this study. After careful consideration, we invite you to submit a revised version of the manuscript.  

We look forward to receiving your revised manuscript.

Kind regards,

Senthilnathan Palaniyandi, Ph.D

Academic Editor

PLOS ONE

Journal Requirements:

2. Please include captions for your Supporting Information files at the end of your manuscript, and update any in-text citations to match accordingly. Please see our Supporting Information guidelines for more information: http://journals.plos.org/plosone/s/supporting-information

Reviewers' comments:

Reviewer's Responses to Questions

**Comments to the Author**

1. Is the manuscript technically sound, and do the data support the conclusions?

Reviewer #1: Partly

Reviewer #2: Yes

2. Has the statistical analysis been performed appropriately and rigorously? 

Reviewer #1: Yes

Reviewer #2: Yes

3. Have the authors made all data underlying the findings in their manuscript fully available?

Reviewer #1: Yes

Reviewer #2: Yes

4. Is the manuscript presented in an intelligible fashion and written in standard English?

Reviewer #1: Yes

Reviewer #2: Yes

5. Review Comments to the Author

Reviewer #1: The authors present data on CMV elispot as a way to monitor CMV specific immune responses. While they have performed significant number of studies in a total of 52 patients, there is just too much heterogeneity to follow though all what they have observed.

There are simply too many variables, the different donors, preparatory regimens, GVHD prophylaxis, types of disease, etc...

these are likely all to influence CMV reactivation. Moreover, it is not clear why relapse disease patients are excluded as they could have had CMV responses.

Also, there is no discussion of what the approach to CMV was... were patients prophylaxed, treated pre emptively or treated at time of CMV disease?

at the end of the manuscript, the only conclusion was that CMV cell mediated immunity is a good end point?

Reviewer #2: This study by Seo et al. examines the recovery of cytomegalovirus-specific cell-mediated immunity following hematopoietic stem cell transplant in pediatric patients. Specifically, the authors compare the recovery profile in recipients of haploidentical cells vs. a group with fully-matched/single mismatch. Although some of the data presented is confirmatory of other studies in different patient populations, this is a nice study that provides more in depth analysis of CMV-specific CMI recovery related to complications and haploidentical donors. I have only a few comments:

1) The authors discuss how in pediatric allogeneic HSCT, more than 80 SFC/2.0 X 10^5 cells can be used as a clinical threshold for showing a protective effect against CMV viremia, while no CMV-related events have been observed pediatric recipients who underwent umbilical cord blood transplantation and had restored CMV-specific CMI >/= 150 SFC/10 X10^6 following HSCT. How did the authors set the definition of recovery and full recovery in their study?

2) Some of the findings suggest the study may not be sufficiently powered to detect differences, for example in

3) The abstract does not mention the haploidentical donor component of the study – this should be added.

4) It would be interesting to plot the haploidentical group vs. total or fully matched/single mismatch in Fig 1. to get a better idea of the impact of the different donors.

6. PLOS authors have the option to publish the peer review history of their article (what does this mean?). If published, this will include your full peer review and any attached files.

Reviewer #1: No

Reviewer #2: No

---

## [Author Response · Author response to Decision Letter 0]

9 Dec 2020

Dear the reviewers,

We appreciate your excellent and informative comments. Your comments were all reflected in the revised manuscript and were indicated in the revised manuscript by using the yellow highlighter tool. Thank you for your deep interest our study again.

Sincerely,

Jina Lee, M.D, Ph.D

Reviewer #1: The authors present data on CMV elispot as a way to monitor CMV specific immune responses. While they have performed significant number of studies in a total of 52 patients, there is just too much heterogenecity to follow though all what they have observed. (There are simply too many variables, the different donors, preparatory regimens, GVHD prophylaxis, types of disease, etc... these are likely all to influence CMV reactivation.)

⇒ Thank you for your kind comment. There is a lot of heterogeneity in our study. However, the reconstitution of T-cell immune response and CMV reactivation after HSCT are affected by various factors. So, we thought that an analysis that considers the interaction between these various factors together might be used more meaningfully in the actual clinical field. We added these to our limitations.

Moreover, it is not clear why relapse disease patients are excluded as they could have had CMV responses.

⇒ We apologize for the lack of clarity. Recipients with a relapse of the underlying disease after HSCT need additional chemotherapy. They may need the use of antiviral drugs or blood transfusions or have the reduction of immune function due to chemotherapy. We thought these factors could influence the recovery of CMV-specific CMI and CMV viremia, so they were excluded from the study. 

Also, there is no discussion of what the approach to CMV was... were patients prophylaxed, treated pre emptively or treated at time of CMV disease?

⇒ Thank you for the comment. The protocol of prophylaxis and pre-emptive therapy for CMV at our institute was added in the revised manuscript as follows;

All recipients who received a HSCT from a matched sibling or unrelated donor did not receive CMV prophylaxis irrespective of CMV serostatus. Instead, as a preemptive treatment, blood CMV viral loads were monitored regularly, and antiviral treatment with either ganciclovir (5 mg/kg/dose every 12 hours) or foscarnet (60 mg/kg/dose every 12 hours) was performed when CMV viremia was detected. This preemptive treatment was discontinued following 2 consecutive negative PCR results.

In recipients with haploidentical HSCT, ganciclovir (5 mg/kg/dose every 24 hours) or foscarnet (60 mg/kg/dose every 24 hours) was used for CMV prophylaxis up to 100 days after engraftment. After that, oral acyclovir was used prophylactically until the first year of transplantation. For CMV viremia, anti-CMV agents were administered for therapeutic purposes. 

At the end of the manuscript, the only conclusion was that CMV cell mediated immunity is a good end point?

⇒ Thanks for letting us know the problem with the conclusion. At the end of the manuscript, we summarized the results of the study and described a proposal to apply the results of our study to actual clinical practice and the benefits of applying it. 

The changes are follows; 

In conclusion, the pre-existence of CMV-specific CMI, donor type, and CMV serostatus before HSCT were the major determinants of general or complete CMV-specific CMI recovery after HSCT, and complete recovery of CMV-specific CMI was important for the prevention of clinically significant CMV infection. Immunologic monitoring using ELISPOT assay before and after HSCT helps in identifying patients with a high risk for CMV infection and in minimizing CMV-related morbidity and mortality. 

Application of this approach could not only reduce the duration of CMV viral load monitoring but also reduce the use of antiviral agents used to control CMV viremia and clinically significant CMV infection and the associated side effects of use of antiviral agents. Further studies among pediatric patients with various serologies and donor types are needed. 

Reviewer #2: This study by Seo et al. examines the recovery of cytomegalovirus-specific cell-mediated immunity following hematopoietic stem cell transplant in pediatric patients. Specifically, the authors compare the recovery profile in recipients of haploidentical cells vs. a group with fully-matched/single mismatch. Although some of the data presented is confirmatory of other studies in different patient populations, this is a nice study that provides more in depth analysis of CMV-specific CMI recovery related to complications and haploidentical donors. I have only a few comments:

1) The authors discuss how in pediatric allogeneic HSCT, more than 80 SFC/2.0 X 10^5 cells can be used as a clinical threshold for showing a protective effect against CMV viremia, while no CMV-related events have been observed pediatric recipients who underwent umbilical cord blood transplantation and had restored CMV-specific CMI >/= 150 SFC/10 X10^6 following HSCT. How did the authors set the definition of recovery and full recovery in their study?

⇒ Thank you for your comments. 

The study by Abate et al., which used ELISPOTs ≥ 80 SFC/2.0 × 105 cells as a clinical threshold for protecting CMV viremia, did not define the recovery of CMV-specific CMI. They analyzed prospectively the CMV-specific T-cell reconstitution in 31 pediatric allogeneic HSCT recipients from sibling or unrelated donors at 30, 60, 90, 120, 180, and 360 days after HSCT. There seems to be no definition of recovery (and full recovery) by conducting research to check immunity on a set day in all patients. They defined more than 80 ELISPOTs as protective factors against CMV viremia based on their ROC curve results. Whereas less than 20 SFC/2.0 × 105 cells was potential risk factor for developing CMV viremia.

In the study by Merindol et al., the recovery of CMV-specific CMI was defined as more than 50 SFC/10 × 105 (10 SCF/2.0 × 105). This was a study on the reconstruction of CMV-specific CMI 3 years after HSCT in 28 pediatric recipients who underwent umbilical cord blood HSCT. No CMV-related events have been observed pediatric recipients who had restored CMV-specific CMI ≥ 150 SFC/10 ×105 (30 SCF/2.0 × 105) following HSCT. 

We added what you commented to explain that sufficient recovery of CMV-specific CMI after transplantation is important in minimizing morbidity and mortality associated with CMV infections in pediatric HSCT recipients

2) Some of the findings suggest the study may not be sufficiently powered to detect differences, for example in

⇒ We have added discussions to explain the similarities and to highlight the differences between our findings and other studies based on several references. We discussed a study of the pediatric haploidentical HSCT recipients and several studies on association with CMV seropositivity of the recipient before HSCT and CMV-specific immunity after HSCT. We also emphasized that the identification of CMV-specific CMI rather than CMV seropositivity of recipients before HSCT type is more meaningful.

3) The abstract does not mention the haploidentical donor component of the study – this should be added.

⇒ We appreciated for your key comment. We added the haploidentical donor component to the abstract. 

4) It would be interesting to plot the haploidentical group vs. total or fully matched/single mismatch in Fig 1. to get a better idea of the impact of the different donors.

⇒ We re-plotted the haploidentical group vs. total or fully matched/single mismatch in Fig 1. to highlight the impact of the different donors as you mentioned.

---

## [Decision Letter · Decision Letter 1]

15 Jan 2021

Immunologic monitoring of cytomegalovirus (CMV) enzyme-linked immune absorbent spot (ELISPOT) for controlling clinically significant CMV infection in pediatric allogeneic hematopoietic stem cell transplant recipients

PONE-D-20-22050R1

Dear Dr. Lee,

We’re pleased to inform you that your manuscript has been judged scientifically suitable for publication and will be formally accepted for publication once it meets all outstanding technical requirements.

Kind regards,

Senthilnathan Palaniyandi, Ph.D

Academic Editor

PLOS ONE

Additional Editor Comments (optional):

Reviewers' comments:

Reviewer's Responses to Questions

**Comments to the Author**

1. If the authors have adequately addressed your comments raised in a previous round of review and you feel that this manuscript is now acceptable for publication, you may indicate that here to bypass the “Comments to the Author” section, enter your conflict of interest statement in the “Confidential to Editor” section, and submit your "Accept" recommendation.

Reviewer #1: All comments have been addressed

2. Is the manuscript technically sound, and do the data support the conclusions?

Reviewer #1: Yes

3. Has the statistical analysis been performed appropriately and rigorously? 

Reviewer #1: Yes

4. Have the authors made all data underlying the findings in their manuscript fully available?

Reviewer #1: Yes

5. Is the manuscript presented in an intelligible fashion and written in standard English?

Reviewer #1: Yes

6. Review Comments to the Author

Reviewer #1: all concerns addressed - no further concerns all concerns addressed - no further concerns no further concerns no further concerns

7. PLOS authors have the option to publish the peer review history of their article (what does this mean?). If published, this will include your full peer review and any attached files.

Reviewer #1: No